# Genetic, Epigenetic and Transcriptome Alterations in Liposarcoma for Target Therapy Selection

**DOI:** 10.3390/cancers16020271

**Published:** 2024-01-08

**Authors:** Ekaterina A. Lesovaya, Timur I. Fetisov, Beniamin Yu. Bokhyan, Varvara P. Maksimova, Evgeny P. Kulikov, Gennady A. Belitsky, Kirill I. Kirsanov, Marianna G. Yakubovskaya

**Affiliations:** 1N.N. Blokhin Russian Cancer Research Center, Ministry of Health of Russia, 24 Kashirskoe Shosse, Moscow 115478, Russia; lesovenok@yandex.ru (E.A.L.); timkatryam@yandex.ru (T.I.F.); beniamin-bokhyan@mail.ru (B.Y.B.); lavarvar@gmail.com (V.P.M.); kkirsanov85@yandex.ru (K.I.K.); 2Faculty of Oncology, I.P. Pavlov Ryazan State Medical University, Ministry of Health of Russia, 9 Vysokovol’tnaya St., Ryazan 390026, Russia; e.kulikov@rzgmu.ru; 3Laboratory of Single Cell Biology, Peoples’ Friendship University of Russia, 6 Miklukho-Maklaya St., Moscow 117198, Russia

**Keywords:** liposarcoma, well-differentiated liposarcoma, dedifferentiated liposarcoma, myxoid/round-cell liposarcoma, pleomorphic liposarcoma, myxoid pleomorphic liposarcoma, molecular genetic abnormalities, epigenetic changes, targeted therapy

## Abstract

**Simple Summary:**

Liposarcoma is the most widespread soft-tissue sarcoma in adults. This review summarizes the molecular genetics and epigenetics of the main liposarcoma subtypes and corresponding aberration in signaling forming the basis for targeted therapy selection. In recent years, specific inhibitors of *CDK4*/6 and *MDM2* and VEGFR/FGFR/PDGFR multi-kinase inhibitors have been proposed for the treatment of liposarcoma.

**Abstract:**

Liposarcoma (LPS) is one of the most common adult soft-tissue sarcomas (STS), characterized by a high diversity of histopathological features as well as to a lesser extent by a spectrum of molecular abnormalities. Current targeted therapies for STS do not include a wide range of drugs and surgical resection is the mainstay of treatment for localized disease in all subtypes, while many LPS patients initially present with or ultimately progress to advanced disease that is either unresectable, metastatic or both. The understanding of the molecular characteristics of liposarcoma subtypes is becoming an important option for the detection of new potential targets and development novel, biology-driven therapies for this disease. Innovative therapies have been introduced and they are currently part of preclinical and clinical studies. In this review, we provide an analysis of the molecular genetics of liposarcoma followed by a discussion of the specific epigenetic changes in these malignancies. Then, we summarize the peculiarities of the key signaling cascades involved in the pathogenesis of the disease and possible novel therapeutic approaches based on a better understanding of subtype-specific disease biology. Although heterogeneity in liposarcoma genetics and phenotype as well as the associated development of resistance to therapy make difficult the introduction of novel therapeutic targets into the clinic, recently a number of targeted therapy drugs were proposed for LPS treatment. The most promising results were shown for *CDK4*/6 and *MDM2* inhibitors as well as for the multi-kinase inhibitors anlotinib and sunitinib.

## 1. Introduction

Liposarcoma (LPS) is a subtype of soft-tissue sarcoma (STS) further divided into five separate groups of malignancies characterized by distinct genetic and molecular aberrations, unique histologic appearance, therapy strategies and overall clinical outcome: well-differentiated liposarcoma (WDLPS), dedifferentiated liposarcoma (DDLSP), myxoid/round-cell liposarcoma (MLPS), pleomorphic liposarcoma (PLPS) and the recently isolated, separate sub-type myxoid pleomorphic liposarcoma (MPLPS), each harboring its own unique features [1]. Although surgical resection and radiotherapy remain the most frequent choices for treatment, chemotherapeutic options are also applied for the treatment of patients with advanced/metastatic clinically unresectable LPS. The specific patterns of disease pathogenesis and progression of each LPS subtype suggest different approaches to improve chemotherapy. An understanding of the genetic and epigenetic abnormalities and corresponding transcriptome changes is critical to the management of liposarcoma and further studies of the mechanisms of liposarcoma pathogenesis.

Well-differentiated (WDLPS) and dedifferentiated (DDLPS) liposarcoma are the most common subtypes of liposarcoma [2,3,4]. WDLPSs are slow-growing malignancies characterized by the presence of adipocytes [2,3,4]. DDLPS is characterized by a higher cellularity and elevated mitosis number [3,4]. DDLPS is a much more aggressive metastatic disease and associated with poor prognosis [5,6,7]. Both subtypes do not have specific age risk factors and usually develop in the retroperitoneum, extremities, paratesticular areas and trunk [3,4,8]. WDLPS and DDLPS are largely resistant to conventional cytotoxic chemotherapy and radiation therapy, and surgery remains the main option [2,3,4].

Myxoid/round-cell (MLPS) liposarcoma is a neoplasm with high cellularity and non-lipogenic, mesenchymal, round- to oval-shaped cells mixed with mature adipocytes [9,10] characterized by a more aggressive disease biology and worse clinical outcome [11]. MLPS is more common in younger patients and predominantly arises in the proximal lower extremities, as opposed to the retroperitoneum [12]. The tumor tends to recur locally and systemically, with a high risk of metastasis to the retroperitoneum, abdomen, chest and trunk [9]. Treatment for MLPS includes surgery and radio- and chemotherapy [13].

Pleomorphic liposarcoma (PLPS) is the most aggressive and histologically non-uniform subtype of liposarcoma. It is a high-grade, aggressive neoplasm consisting of pleomorphic lipoblasts and occasional multinucleated giant cells [2,3,4,9]. The median age of the patients is 55-65 years old and they most commonly present with disease in the lower extremities. These malignancies are highly resistant to all current treatment modalities [14,15].

Myxoid pleomorphic liposarcoma (MPLPS) is an exceedingly rare adipocytic malignancy developing in the mediastinum, followed by the limbs and the head and neck region. Morphologically, MPLPS shows features of both myxoid and pleomorphic liposarcoma with aggressive clinical behavior, including fast tumor growth and early metastasis to the lungs, bone and soft tissues [1]. Genetic and epigenetic results suggest a possible link with conventional pleomorphic liposarcoma [16].

In practice, distinguishing one liposarcoma subtype from another is rather challenging. Molecular studies should follow the histologic examination for more accuracy in diagnostics and optimal disease therapy course or enrollment into clinical trials.

## 2. Molecular Genetic Abnormalities and Corresponding Transcriptome Changes Specific to Liposarcomas and Their Possible Role as Therapeutic Targets

A number of the genetic abnormalities are specific to the whole set of LPSs: TOP2A, PTK7 and CHEK1 were overexpressed in 140 cases of liposarcoma [17]; point mutations in CTNNB1, CDH1, FBXW7 and EPHA1, C-MET and EGFR amplification and increased expression of C-KIT, EGFR, PD-L1 and PD-1 also represent potential oncogenic events in liposarcoma cells [18]. Loss of estrogen receptor expression may be involved in the pathogenesis of liposarcoma through an unknown mechanism [19]. The transcription factor TBX3, a critical developmental regulator, was shown to have a role as an oncogene/motogene in liposarcoma [20].

Additionally, the specific genetic alterations found were specific to several subtypes of liposarcoma.

### 2.1. Well-Differentiated Liposarcoma (WDLPS) and Dedifferentiated Liposarcoma (DDLSP): 12q13-15-Associated Chromosomal Aberrations as Major Driver of Pathogenesis

WDLPS and DDLPS usually share the same genetic aberration, represented by the distinctive ring and/or giant marker chromosomes from the 12q13-15 segment (Table 1 and [21]). This chromosome region bears more than 350 genes, including multiple proliferative genes [22]. In particular, the most common overamplified genes in WDLPS/DDLPS are the member of the High-Mobility Group A (HMGA) gene family *HMGA2*, encoding the transcriptional factor modulating the chromatin structure in the nucleus [21,23]; *CDK4*, gene of cyclin-dependent kinase 4 [24]; pro-proliferative genes from the *JUN* family [25]; and mouse double minute 2 (*MDM2*), encoding a well-studied inhibitor of the p53 tumor suppressor [24,26,27]. These genes are well-studied in the context of WDLPS/DDLSP and reveal several correlations with the type of malignancy, location, grade, node involvement, distant metastasis and recurrence-free survival [25]. Other genes frequently amplified within the 12q13-15 amplicon include tetraspanin 31 (TSPAN31), a gene with possible role in the proliferation, migration and inhibition of apoptosis [28,29]. YEATS4, a proliferative gene, and CPM, encoding carboxypeptidase M, a proteolytic enzyme inducing cleavage activation of growth factors, are genes commonly amplified within 12q13-15 that have been implicated in dedifferentiation [30]. FRS2, E2F1 and CDKN2A are also among the most upregulated genes in DDLPS and WDLPS [26,31]. Notable deletions were found in chromosome 1p (RUNX3, ARID1A), chromosome 11q (ATM, CHEK1) and chromosome 13q14.2 (MIR15A, MIR16-1) [30]. It was also demonstrated for WDLPS/DDLPS without the *CDK4* amplification that an alteration in the CDKN2A/CDKN2B/*CDK4*/CCND1 pathway is present in almost all cases without *CDK4* amplification and may play a pivotal role in oncogenesis [32].

In WDLPS/DDLPS, the molecular features of malignancies may vary between subtypes. In particular, insulinoma-associated protein 1 (INSM1) is a specific biomarker for neuroendocrine cancers, but its expression is also detected in liposarcomas. Moreover, INSM1 expression in WDLPS was significantly higher than in adipocytes and DDLPS cells. Significant differences in the expression of INSM1 in WDLPS and DDLPS may assist in the diagnosis, enriching the diagnostic index system of mesenchymal cancers [43]. Additional chromosomal abnormalities, more exclusive for DDLPS than for WDLPS, are recurrent amplifications of 1p32 and 6q23, in particular, overexpression of ASK1, DDR2, ERBB3, STAT6, FGFR1, MAP3K5, LGR5, MCL1, CALR, AQP7, ACACB, FZD4, GPD1, LEP and ROS1 [21,44,45,46]. Another set of core genes in DDLPS identified as significantly enriched in microarray profiling generated from DDLPS and normal fat controls include APP, *MDM2*, CDK1, PCNA, TKT, *CDK4*, CDC20, BUB1B, BARD1, ADRB2, LGALS3, CAV1, CCNA2 and CDKN2A. The pathways identified as enriched in DDLPS are the pyruvate pathway, cell cycle genes and molecular mechanisms associated with the DDLPS pathway and PPAR signaling pathway [47]. CTDSP1/2-DNM3OS fusion genes were identified in a subset of DDLPS tumors by integrating exome and RNA sequencing data [48].

Several genes located at 19p13.1-13.2 were highly expressed in DDLPS, including genes encoding CRT, the inhibitor of adipocyte differentiation, and CD47, tightly associated with malignant transformation [18]. The expression of the E3-ubiqutin ligase gene SIAH2 in DDLPS tumor-associated macrophages and other stromal cells indicates that SIAH2 expression may serve as a molecular marker distinguishing between DDLPS and WDLPS, but more complete evaluation of the role of SIAH2 in the DDLPS phenotype is limited by the availability of fresh tissues from these rare cancers [49]. In a study of the role of α-thalassemia/mental retardation syndrome X-linked (ATRX) or death domain-associated protein 6 (DAXX) gene expression in telomerase activation and alternative lengthening of telomeres, a 100% correlation was demonstrated between ATRX or DAXX and alternative telomere lengthening in DDLPS. It was also correlated with poor survival, suggesting the prognostic role of ATRX and DAXX in DDLPS [50]. Expression of the PD-1 gene, encoding the differentiation marker of the immune cells, was particularly high in DDLPS [51]. Another study reported a correlation between high expression of the centromere protein F (CENPF) gene and worse survival of DDLPS patients, therefore suggesting CENPF as a malignant indicator of tumor immune infiltration-related survival [52]. Rare DDLPS-specific alterations are mutations in the fibroblast growth factors FGFR1, FGFR2, FGFR3 and FGFR4, as well as in FGFR substrate 2 (FRS2), characterized by a poor prognosis [18,53,54,55].

For WDLPS pathogenesis, a second amplicon originating from 10p11-14 is described containing 62 genes, including oncogenes such as MLLT10, previously described in chimeric fusion with MLL in leukemias, NEBL and BMI1 [22]. SORBS1, KRT8 and MT1G are among the top downregulated genes in WDLPS and DDLPS [31]. MT1G was previously reported to be a tumor suppressor and was silenced in hepatocellular carcinoma [56]. Low SORBS1 expression is associated with promotion of invasion and metastasis as well as an overall poor prognosis in breast cancer [57]. CCAAT/enhancer binding protein (CEBPA) and PPAR-γ are reported to be downregulated in DD/WDLPS but more frequently in DDLPS [18].

### 2.2. Myxoid and Round-Cell Liposarcoma (MLPS): DNA Damage-Associated Gene CHOP and Its Translocation Partners

MLPS is characterized by unique chromosome rearrangements, namely, t(12;16) (q13;p11), that result in the *FUS-CHOP* (*FUS-DDIT3*) gene fusion in more than 95% of cases or the rarer translocation t(12;22) (q13;q12), leading to the formation of the *EWS*-CHOP oncogene in 5% of malignancies [18]. The gene CHOP encodes a growth arrest and DNA-damage inducible member of the C/EBP family of transcription factors, regulates adipogenesis and assists in growth arrest, but loses the function after the rearrangement and stimulates proliferation [58]. The CHOP translocation partners include a TLS gene of nuclear RNA-binding protein and an *EWS* gene with great similarity to TLS, whose protein product is involved in the development of a wide variety of cancers, including Ewing’s sarcoma, melanoma and several neuroendocrine cancers [33]. Interestingly, the breakage in the introns of the CHOP gene with further formation of chimeric genes suggests the presence of a characteristic sequence in the breakpoint regions, including the mobile element Alu and palindromic oligomer sequences [34]. To date, eleven *FUS-CHOP* and five *EWS-CHOP* chimeric genes have been described [35]. The corresponding aberrant proteins interfere with normal adipocyte differentiation and are involved in the activation of several tyrosine kinase receptor pathways including MET, RET, IGFR, AXL, EGFR, PI3K/Akt and VEGFR2 specifically for round-cell liposarcoma [18]. Activating mutations or amplification of PIK3CA, P110α catalytic subunit mutations of PI3K are seen in approximately 15% of MLPS and are associated with a poor prognosis, whereas PTEN deletion has also been described [18,21]. MLPSs are also characterized by a high frequency of hotspot mutations (C228T or C250T) in the promoter region of telomerase reverse transcriptase (TERT), which encodes the TERT protein responsible for telomerase reactivation [36,59]. TERT mutation is associated with a poor prognosis in MLPS; however, it could not be depicted as a prognostic factor. Thus, in a retrospective study on 83 primary MLPS tumor samples, TERT hotspot mutations were observed in 77% of cases, but aberrant telomere lengthening was not detected. Furthermore, TERT promoter hotspot mutations did not correlate with patient survival [60], in contrast with ATRX/DAXX overexpression and alternative telomere lengthening in DDLPS [50]. Gene expression studies have reported the specific expression of CTAG1B, CTAG2, MAGEA9 and PRAME in myxoid and round-cell liposarcoma [61]. High expression of the CHSY1 gene encoding surface glycosaminoglycan could be an additional marker of malignant pathologic grade and poor clinical prognosis in soft-tissue sarcomas with myxoid substance [62]. STAT6 can also be overexpressed in myxoid liposarcoma [46].

### 2.3. Pleomorphic Liposarcoma (PLPS) and Myxoid Pleomorphic Liposarcoma (MPLPS): Complex Karyotype and Poor Prognosis

PLPS and MPLPS are usually characterized by complex karyotypic aberrations without specific genetic alterations. Comparative genomic hybridization analyses showed gains of 1p, 1q21-q32, 2q, 3p, 3q, 5p12-p15, 5q, 6p21, 7p, 7q22, 8q, 10q, 12q12-q24, 13q, 14q, 15q, 17p, 17q, 18p, 18q12, 19p12, 19q13, 20q, 22q and Xq21-q27 and losses of 1q, 2q, 3p, 4q, 10q, 11q, 12p13, 13q14, 13q21-qter, 14q23-24, 16q22, 17p13, 17q11.2 and 22q13 [41,42]. TP53 mutations are observed in 60% of PLPS patients [26], deletion of 13q14.2-5 (containing the tumor-suppressor gene RB1) in up to 50% [21] and loss of tumor the suppressor-gene NF1 in 5% of patients [30]. In a study of 155 patients diagnosed with PLPS, increased expression of PPARγ (adipogenic marker), BCL2 and survivin (survival factors), VEGF (angiogenic factor), MMP2 metalloprotease and other biomarkers was revealed [15,18]. Amplification of δ catenin on 5p and deregulation of genes involved in adipogenesis (CEBPA on 19q, EP300 on 22q13) associated with the promotion of metastasis and loss of adipocyte differentiation are also observed [41].

### 2.4. Conclusion on Liposarcoma Genetics

To sum up, some genetic alterations with oncogenic potential are described for all subtypes of liposarcoma. The most frequent WDLPS and DDLPS genetic aberration is represented by the 12q13-15 segment rearrangements, affecting the expression of more than 60 genes, including pro-proliferative ones. An additional frequent transcriptome abnormality for DDLPS is represented by the overexpression of several genes located at 19p13.1-13.2. Telomerase activation and alternative lengthening of telomeres were also demonstrated for DDLPS. Moreover, high expression of PD-1 was found in DDLPS tumor-associated macrophages. For WDLPS, a second amplicon originating from 10p11-14 is described. Several genes, including SORBS1, KRT8 and MT1G, are downregulated in WDLPS and DDLPS. MLPS is characterized by the translocation (12;16) (q13;p11), resulting in the *FUS-CHOP* gene fusion and affecting adipocyte differentiation and the activation of tyrosine kinases MET, RET, IGFR, AXL, EGFR, PI3K/Akt and VEGFR2. Additionally, overexpression of CTAG1B, CTAG2, MAGEA9, PRAME and CHSY1 was described in MLPS. As concerns tumor-suppressor genes, PTEN deletion is also not uncommon in this type of liposarcoma. PLPS and MPLPS are characterized by complex karyotype and simultaneous aberrations simultaneously with P53 mutations and the deletion of 13q14.2-5, including RB1.

## 3. Epigenetic Markers of Liposarcoma

Epigenetic regulation of gene expression occurs on multiple levels, including DNA methylation, histone mutations and modification, chromatin structure alterations and remodeling, the formation of alternative DNA structures as well as transcription regulation by specific subsets of long non-coding RNA (lncRNA) and miRNA [45,63]. Novel manners of cell communication and genetic exchange such as exosomes, macrovesicle, and apoptotic bodies containing miRNAs with LPS-relevant functions involve adjacent and distant recipient cells and add complexity to this situation [45]. It has to be noted that studies of LPS epigenetics have not been reported all LPS subtypes, and the use of epigenetic modulators in therapy for liposarcoma should develop a stronger basis. WDLPS and DDLPS are already characterized by a multi-component landscape of histone modifications and histone-modifying enzymes as well as by a specific miRNA profile. In contrast, there are no data describing the epigenetic changes in PLPS and MPLPS. Nevertheless, specific miRNAs, in particular, miR-215-5p, was shown to promote *MDM2* expression in liposarcoma without specificity to a certain subtype. In addition, it was found to promote cell proliferation, inhibit apoptosis, promote cell cycle progression and promote cell invasion and migration. Therefore, miR-215-5p could be considered a novel therapeutic target in liposarcoma [64]. Hypermethylation of H3K4me3 and H3K9me3 was found in a study of patient-derived xenografts from upper-abdominal soft-tissue liposarcoma. This epigenetic feature may be related to methionine addiction, a fundamental hallmark of cancer, termed the Hoffman effect [65]. The over-methylation of these histone marks requires excess methionine in the form of S-adenosylmethionine and may, at least in part, account for the excess methionine required by cancer cells [66]. Liposarcoma subtypes have their unique genetic and clinical characteristics, undoubtedly cross-talking with the epigenetic features of specific malignancies. Below, we review the current and potential future epigenetic prognostic markers and/or therapy targets.

### 3.1. Mutations in the Genes of Epigenetic Regulators and the Whole Set of Differentially Expressed miRNAs in WDLPS and DDLSP

In DDLPS, specific methylation profiles correlate with clinical outcomes [67,68]. In many cases, promoter elements are hypomethylated, while enhancers and coding sequences are hypermethylated, although the net consequences on transcription in vivo are not entirely predictable [67].

Mutations in genes of epigenetic regulators, specifically in histone deacetylase 1 HDAC1, were demonstrated for DDLPS, but the significance of HDAC1 mutations in DLPS remains to be fully defined at the biochemical level [69]. A comparative analysis of epigenetic modifications and the DNA methylation level in DDLPS identified 833 differentially methylated regions affecting the promoters of 677 genes [70]. Significant tumor-specific promoter methylation associated with downregulation was found in KLF4 and CEBPA, encoding two transcription factors associated with adipocyte differentiation. KLF4 regulates CEBPA, and loss of expression of these factors is considered to be tumorigenic [70]. A study of DNA methylation status and gene expression levels in a large and representative cohort of 80 untreated, primary high-grade sarcomas composed of eight subtypes revealed the prognostic value of DNA hypermethylation of CpG sites in the CDKN2A gene in PLPS and DDLPS [71]. p16INK4a gene promoter hypermethylation is considered to be a potential marker for DDLPS but not for WDLPS [72].

More than 40 miRNAs were found to be differentially expressed in DDLPS and WDLPS among themselves as well as in comparison to normal fat [18]. One of the most frequently upregulated miRNAs in DDLPS is miR-155, involved in malignization via the regulation of casein kinase 1α (CK1α), which results in the activation of the β-catenin pathway [73]. β-catenin and its downstream effector cyclin D1 were found to be overexpressed in all human DDLPS cell lines compared with preadipocytes and adipocytes and were also shown to induce DDLPS cell proliferation and cell cycle progression [73,74]. Knockdown of miR-155 inhibited DDLPS cell proliferation, decreased colony formation, induced cell cycle arrest in vitro and blocked tumor growth in xenografts [75]. MiR-193 family members were found to be downregulated in DDLPS compared with normal fat, and miR-193 expression is considered a favorable prognostic factor in WDLPS/DDLPS [76], as well as a therapeutic approach, as miR-193 targets PDGFRβ, SMAD4 and YAP1, belonging to strongly interacting pathways (focal adhesion, TGFβ and Hippo, respectively) [77]. Interestingly, the expression of miR-193b in liposarcoma cells was downregulated by promoter methylation, resulting at least in part from increased expression of the DNA methyltransferase DNMT1 in WDLS/DDLS, which leads researchers to also consider the immediate implication of demethylation agents for therapeutic exploration [76,78]. MiR-143, which is abundant in normal adipose tissue, was found to be underexpressed in WDLPS, and its expression decreased further as the tumor progressed to DDLPS. The signaling targets of miRNA-143 include BCL2, TOP2A and PLK1 [79]. The role of miR-145 and miR-451 in the suppression of tumor growth was demonstrated for DDLPS, as well as the tumor-promoting role of miR-26a in DDLPS/WDLPS [80]. Loss of miR-133a expression induces a metabolic shift due to a reduction in oxidative metabolism favoring a Warburg effect in DDLPS [81].

In a study of the expression of 1888 miRNAs in 25 human liposarcoma samples, a DDLPS-specific downregulated subset of miRNAs was described, including miR-144, miR-451, miR-29b-2, miR-365, miR29b miR-499-5b, miR-486-5p and miR-551 [82]. Further, the role of the miRNAs miR-133a, miR-199a-3p, miR25-3p and miR-92a-3p was investigated in DDLPS progression, but a correlation between the expression of miRNAs and tumor viability was shown only for miR-199a-3p [18]. MiR-133, miR-1 and miR-206 were significantly underexpressed in WDLPS and may function as tumor suppressors, as described in muscle-relevant rhabdomyosarcomas [83]. Tan et al. described other specific subsets of miRNAs in DDLPS and WDLPS: they confirmed the upregulation of miR-214-3p, miR-199a, miR-21-3p and miR-21-5p and downregulation of miR-10b, miR-126-3p, miR-126-5p, miR-143-3p, miR-143-5p, miR-145-5p and miR-193b-3p in WDLPS/DDLPS compared to benign lipoma [84]. MiR-3613-3p is upregulated in DDLPS patients and may serve as a potential specific biomarker for dedifferentiated liposarcoma [85]. The analysis of tissue and serum miRNA expression in DDLPS identified miR-1246, -4532, -4454, -619-5p and -6126 as biomarkers for DDLPS [86].

### 3.2. MLPS: FUS-CHOP-Associated Chromatin Remodeling and Changes in Specific miRNA Expression

The specific methylation profile of the 12q13-q14 region in MLPS with t(12;16) (q13;p11) translocation has been described [38]. Epigenetic analyses showed that 45% of myxoid/round-cell liposarcomas were CpG-methylated at the APC locus and had reduced APC expression [39]. Increases in expression of CDKN2A, MGMT, RASSF1A, MST1 and MST2 were also found to be epigenetically regulated by the DNA methylation level [40].

Chromatin remodeling plays a role in MLPS through interactions between *FUS-DDIT3* and components of the subfamily of ATP-dependent chromatin remodeling complexes SWI/SNF and polycomb repressive complex 2 PRC2 [87,88,89]. The histone code reader Spindlin1 (SPIN1) was shown to impair proliferation and increase apoptosis of liposarcoma cells in vitro and in xenograft mouse models. Using signaling pathway, genome-wide chromatin binding and transcriptome analyses, Franz et al. found that SPIN1 directly enhances the expression of GDNF, an activator of the RET signaling pathway, in cooperation with the transcription factor MAZ. Importantly, a mutation of SPIN1 within the reader domain interfering with chromatin binding reduces liposarcoma cell proliferation and survival. These data suggest SPIN1 as a novel target for chromatin-associated small-molecule inhibitors [90]. In a study of integral DNA methylation patterns in liposarcoma samples, it was demonstrated that ALDH1A3 was the most hypermethylated and downregulated gene for MLPS compared to normal fat [71]. ALDH1A3 is a member of the aldehyde dehydrogenase family with 19 isoenzymes that potentially plays a role in the detoxification of aldehydes in alcohol metabolism and lipid peroxidation [91]. High ALDH1 activity in sarcoma cell lines is associated with an increase in proliferation [92]. The EFEMP1 gene encoding fibulin-3, a member of the extracellular matrix glycoprotein family associated with lymph node metastasis, vascular invasion and poor prognosis, was also found to be hypermethylated and downregulated in MLPS compared to normal fat [71,93,94,95].

A lesser extent of specific miRNAs is described for MLPS. Thus, microRNA-135b (miR-135b) is described as a key regulator of the malignancy, promoting MLPS metastasis in vivo through the direct suppression of thrombospondin 2 (THBS2) and following an increase in the total amount of MMP2 [37]. Another study demonstrated the role of high expression of miR-9, miR-9* and miR-31 in the progression and metastasis of MLPS [96]. It was demonstrated that miR-486 expression was repressed in TLS-CHOP-expressing MLS tissues, so downregulation of miR-486 may be an important process for MLPS development [97].

### 3.3. Conclusions of Liposarcoma Epigenetics

In conclusion to the epigenetic section, it should be noted that almost all liposarcoma subtypes accumulate a number of epigenetic alterations, which could be considered possible therapy targets. In particular, the whole pool of target miRNAs in DDLPS, WDLPS and MLPS is described as drivers/markers of pathogenesis and are under extensive investigation. Hypermetylation of H3K4me3 and H3K9me3 in the abovementioned LPS subtypes may lead to the hyperexpression of cell cycle regulators and a decrease in the expression of tumor-suppressor genes such as APC. No data on epigenetic-specific features of PLPS and MPLPS have been described in the literature.

## 4. Changes in Signaling and Therapeutic Approaches

Treatment of liposarcoma typically involves surgery and radiation therapy, while the use of classic cytostatic treatment and targeted therapy frequently lead to the development of resistance at the advanced disease stage. However, multiple translational studies of novel therapies target various genetic and molecular aberrations in different subtypes of liposarcoma. In particular, WDLPS/DDLPS-specific aberrations in the 12q13-15 amplicon leading to the amplification of *MDM2* and *CDK4* and MLPS-specific *FUS-DDIT3*/*EWSR1-DDIT3* fusion represent potential therapeutic candidates. Moreover, several low-molecular-weight multi-kinase inhibitors targeting MET, AXL, IGF1R, EGFR, VEGFR2 and PDGFR-β could be effective in the types of liposarcoma characterized by abnormalities in PI3K/Akt/mTOR signaling and the associated deregulation of other cascades [98,99,100,101,102].

### 4.1. MDM2/p53 and CDK4 Signaling Aberrations as Well as Activation Mutations in Multiple Growth Factors in WDLPS and DDLPS

As described in the section “Molecular genetic abnormalities”, *MDM2* and *CDK4* are frequently co-amplified in WDLPS and DDLPS [103]. Amplification of *MDM2* results in the inactivation of p53, and *CDK4* amplification leads to cell cycle progression [53,104]. Both alterations can be targeted by specific inhibitors (*MDM2* antagonists RG7388 and Nutlin 3A (RG7112); *CDK4*/6 inhibitors palbociclib, ribociclib, abemaciclib and TQB3616) in experimental and clinical trials either used individually or in combination, especially in the therapy of DDLPS [98,100,103,105,106,107,108,109,110,111,112]. However, it has to be noted that combinations of *MDM2* and *CDK4* inhibition in DDLPS should be thoroughly investigated in clinical studies due to the possible combined toxicities of these drugs [106].

The orally bioavailable selective inhibitor of nuclear export selinexor has been demonstrated to have preclinical activity in various cancer types and is currently in phase I and II clinical trials for advanced cancers. It was shown in vitro that selinexor induces G1-arrest in liposarcoma cell lines with *MDM2* and *CDK4* amplification by increasing the protein level of p53 and p21, indicating a post-transcriptional effect. These results justify the exploration of selinexor in clinical trials targeting various sarcoma subtypes [113].

*MDM2* inactivates p53 in a phosphorylated form. Dephosphorylation and depletion of *MDM2* by the inhibitor of HDAC resulted in increased apoptosis, anti-proliferative effects and cell cycle arrest in liposarcoma cell lines, warranting further evaluation of HDACi as a therapeutic option in *MDM2*-amplified LPS [114]. Another epigenetic approach to the treatment of DDLPS/WDLPS is the inhibition of specific miRNAs. Thus, promotion of *MDM2* expression, cell proliferation and invasion of the liposarcoma SW-872 cell line as well as inhibition of apoptosis by miR-215-5p is described in the literature. Targeting miR-215-5p may be a novel therapeutic strategy for the treatment of liposarcoma [64].

Following *MDM2*/P53 signaling, these molecules are linked to PTEN and the PI3K/Akt/mTOR pathway, regulating the pro-apoptotic and anti-apoptotic signals. Specifically, *MDM2* could be stabilized by Akt-mediated phosphorylation, and, in turn, inhibit PI3K/Akt activity via prevention of the nuclear localization of the tumor suppressor REST [104]. PTEN expression in patient samples correlates with poor survival [115]. The PTEN-controlled PI3K/Akt/mTOR pathway could be a therapeutic target for DDLPS, as PTEN protects p53 from *MDM2*-mediated degradation. Together with the inhibition of PI3K/Akt/mTOR signaling, it can augment P53-mediated apoptosis, as was demonstrated in multiple studies in vitro and in vivo [104]. PI3K/Akt/mTOR inhibitors, for example, BEZ235, could be an option for combined treatment of WDLPS/DDLPS [104,116]. Further, downstream Akt targets c-Jun N-terminal kinase (JNK) from the mitogen-activated protein kinase (MAPK) family and this cross-talk may be useful in the development of therapy approaches [117]. However, in phase II trials, the multi-kinase, dual-action inhibitor sorafenib demonstrated a lack of significant clinical efficacy in liposarcoma treatment [118].

Mutational events in the fibroblast growth factor receptors FGFR1, FGFR2, FGFR3 and FGFR4 and the FGFR substrate FRS2 suggest that FGFR signaling plays a role in the pathogenesis of liposarcoma, especially in the development of high-grade DDLPS [18,53]. Moreover, a combination of the FGFR inhibitors erdafitinib and NVP-BGJ398 together with the *MDM2* antagonist RG7388 was shown to be a promising strategy for the treatment of DDLPS and needs further investigation in clinical trials [55,119].

In addition to the TP53 and RB signaling pathways, other pathways may be involved in the dedifferentiation process from WDLPS to DDLPS, including mitogenic and motogenic Wnt and Hedgehog signaling cascades, as well as Notch signaling regulation the differentiation. Besides their overall tumorigenic properties, a specific association of Wnt, Hedgehog and Notch activation with malignant transformation was demonstrated in DDLPS and WDLPS. However, there is no clear evidence for a role of this pathway in regulating tumor progression and the dedifferentiation process [18]. Another Akt downstream target is insulin-like growth factor 1 receptor (IGF1R). IGF1R inhibitors are early-stage therapeutics, and their potential synergistic effect in combination with *CDK4*/6 inhibitors were predicted in silico and proved in vitro [120]. Several receptors, including MET, PDGFR, AXL, VEGFR and EGFR as well as Aurora kinase family proteins are overexpressed in WDLPS/DDLPS. All these receptors may act as targets and have already available small-molecule inhibitors, and some of them have already demonstrated anti-proliferative and proapoptotic effects in liposarcoma cells [18,45]. Noteworthily, the multi-kinase angiogenesis inhibitor anlotinib demonstrated in preclinical and clinical studies a higher efficacy compared to the multi-kinase inhibitors sorafenib, sunitinib and nintedatinib [102]. A phase II trial showed the promising efficacy and acceptable toxicity of anlotinib as maintenance treatment after first-line anthracycline-based chemotherapy [121,122].

Peroxisome proliferator-activated receptors (PPAR) regulate normal adipocyte differentiation. PPAR-gamma is regulated by c-*JUN* and induces the differentiation of normal preadipocytes. Hyperactivation of c-*JUN* blocks differentiation and may contribute to malignant transformation. PPAR-gamma agonists revealed antitumor activity in vitro in liposarcoma cell lines. In this sense, PPAR-gamma represents an attractive target, particularly for DDLPS, MLPS, and in some cases, PLS as a mechanism to revert these subtypes to a well-differentiated phenotype. However, clinical trials with the PPAR-gamma ligands demonstrated mixed results. The PPAR-gamma agonist troglitazone was used for the treatment of patients with advanced liposarcoma and demonstrated expression of several mRNA transcripts characteristic of adipocytic differentiation and a marked reduction in cancer cell proliferation [123]. Two other clinical trials with rosiglitazone and efatutazone demonstrated mixed results [124,125].

### 4.2. FUS-CHOP-Associated Abnormalities of PI3K/Akt/mTOR and Other Proliferative Signaling in MLPS

Fusion proteins from the chimeric oncogenes *FUS-DDIT3* and *EWS1R-DDIT3* act as aberrant transcription factors and may affect many signaling molecules. Thus, gene expression studies in MLPS have identified the recurrent upregulation of MET, RET, IGF-IR and PIK3CA, suggesting that these genes to be downstream targets of MLPS-specific fusion proteins [104]. Mutations in the PI3K catalytic subunit, IGF1R expression, amplification and mutations in PIK3Ca and loss of PTEN are reported in 12–18% of cases, therefore affecting multiple PTEN and PI3K/Akt/mTOR downstream genes [18,104]. The use of IGF-IR/PI3K/Akt/mTOR inhibitors in therapy for MLPS has a therapeutic potential and is currently under investigation. More specifically, treatment of myxoid liposarcoma cell lines in vitro and xenograft-bearing mice in vivo with several IGF-IR and PI3K/Akt/mTOR inhibitors resulted in significant growth inhibition [126,127]. One of the mechanisms of tumor heterogeneity and oncogenic potential maintenance is the phosphorylation of Interacts With SUPT6H (IWS1), a regulator of histone activity, by AKT. These findings support the use of the AKT/IWS1 axis as a novel prognostic factor and potential therapeutic target in liposarcoma therapy [128].

Another microarray analysis revealed overexpression of FGFR2 and other members of the FGF/FGFR family and the efficacy of the FGFR inhibitors PD173074, TKI258 (dovitinib) and BGJ398 in experiments in vitro [129]. In addition, the FUS-DDIT3 protein induces increased expression of the CAAT/enhancer-binding protein (C/EBP) and nuclear factor NFKBIZ, a member of the NF-kB family, colocalizing with FUS-DDIT3 [130]. A study of the kinome of cell lines and primary cell cultures from patients with metastatic myxoid liposarcoma revealed the activation of the kinase set associated with activation of the atypical nuclear factor-kappaB and the Src pathways. Moreover, in vitro NF-kB suppression by Casein kinase II inhibitor TBB and Src inhibition using dasatinib decreased cancer cell viability and offered potential therapeutic strategies for myxoid liposarcoma patients with advanced disease [131].

The Hippo pathway effector and transcriptional co-regulator YAP1 was shown to be a downstream target of FUS-DDIT3. In vitro studies demonstrated that FUS-DDIT3-driven IGF-IR/PI3K/AKT signaling promotes stability and nuclear accumulation of YAP1 via deregulation of the Hippo pathway. Gene expression profiling revealed gene signatures related to proliferation, cell cycle progression, apoptosis and adipogenesis. Therefore, FUS-DDIT3 involves IGF-IR/PI3K/AKT signals via Hippo/YAP1, and YAP1 may be an immunohistochemical marker for MLPS diagnostics. Moreover, these findings provide a rationale for the development of low-molecular-weight inhibitors of key components in Hippo/YAP1 signaling [132,133,134].

FUS-DDIT3-associated malignant transformation of adipocytes resulted in elevated levels of STAT3 and phosphorylated STAT3, suggesting the involvement of JAK/STAT signaling in the pathogenesis of MLPS [135]. Several inhibitors targeting JAK and GSK-3 caused downregulation of FUS-DDIT3 in vitro and reduced cell proliferation [136].

The components of the VEGF signaling pathway FLT1, PGF, VEGFA and VEGFB were shown to be indirect targets of FUS-DDIT3 in vitro. This could be a consequence of the ability of FUS-DDIT3 to reprogram primary adipocytes to a liposarcoma-like phenotype [137]. One case is reported in the literature of a 68-year-old Chinese woman initially diagnosed with advanced multiple intra-abdominal and pelvic round-cell liposarcomas who responded to therapy with the VEGFR2 inhibitor apatinib [138]. Further clinical trials are needed to confirm the efficacy and safety of VEFGR inhibitors in the treatment of MLPS.

MLPSs, like some other malignancies associated with chromosomal translocations resulting in expression of a fusion protein, are more responsive to trabectedin than other sarcoma types. Trabectedin does not act as an inhibitor of specific hyperactivated/overexpressed proteins; it binds covalently to the exocyclic amino group of guanines in the DNA minor groove, competing with the fusion protein and preventing its transcriptional activity. Several clinical studies have demonstrated the efficacy and favorable safety profile of trabectedin [18,112].

### 4.3. PLPS and MPLPS: No Specific Targets

No data on specific features in signaling and targeted treatments of pleomorphic liposarcoma are described in the literature. It is the rarest type of liposarcoma with poor prognosis, and its therapy involves mainly surgical management and the application of radiation. In addition, PLPS and MPLPS may respond to a doxorubicin and ifosfamide combination; trabectedin and eribulin are also options for advanced disease. A reduction in the primary tumor and the eradication of lung metastasis were reported in a clinical case of combined PLPS treatment with the multi-kinase inhibitor pazopanib, eribulin and dacarbazine [139]. Significant work remains to be done to develop novel therapies for this disease. To date, most studies have failed to identify targetable aberrations and have noted only consistent losses in p53 and Rb pathway proteins [18,140,141,142]

### 4.4. Perspectives of Targeted Therapy for Liposarcomas

Currently, CDK 4/6 and *MDM2* amplifications present the prospective targets for LPS therapy, and the efficacy of *CDK4*/6 and *MDM2* inhibitors was proved in clinical trials on WDLPS and DDLPS (Table 2). Multi-kinase inhibitors including the well-studied sunitinib and the most recent, anlopanib, demonstrated mixed results, suggesting the necessity of further studies. To date, their combination with standard radiotherapy and conventional cytostatic approaches is still required until chemoresistance to the standard therapy appears. As nowadays chemoresistance prediction based on molecular genetics analysis seems insufficient, the development of experimental approaches for testing it ex vivo and in vitro may be useful for the exclusion of potentially ineffective targeted therapy courses and the choice of more promising treatment strategies.

## 5. Conclusions/Future Direction in Therapy

Although multiple key genetic and epigenetic aberrations in liposarcoma have been explored, only a few of them have given rise to novel targeted therapy courses. The heterogeneity and very variable percentage of genetic and epigenetic abnormalities lead to an insufficient understanding of the complex signaling changes enabling tumor progression and high chance of development of tumor resistance. Notably, the reviewed data on specific genetic abnormalities taken together present a cluster of genetically characterized liposarcomas that may be considered for targeted therapies. The results of clinical trials of *CDK4* and *MDM2* inhibitors in the case of WDLPS/DDLPS and multi-kinase inhibitors targeting the FUS-CHOP downstream proteins seem promising. In many other cases, the complexity of sarcoma genetics could impede the diagnostics and may lead to tumor resistance and a poor prognosis. However, the combination with standard radiotherapy and conventional cytostatic approaches is still required until chemoresistance to the standard therapy appears. As nowadays chemoresistance prediction based on molecular genetics analysis seems insufficient, the development of experimental approaches for testing it ex vivo and in vitro may be useful for the exclusion of potentially ineffective targeted therapy courses and the choice of more promising treatment strategies. Overall, further data accumulation is required in the field of LPS molecular pathogenesis as well as in clinical trials of specific inhibitors, as the first target therapy applications gave rather promising results. A better understanding of the distinct genetic and molecular aberrations of liposarcoma subtypes may allow the development of several novel biology-driven therapies based on the specific molecular genetic profile of the disease.

## Figures and Tables

**Table 1 cancers-16-00271-t001:** The most frequent genetic and epigenetic aberrations in LPS.

LPS Subtype	Cytogenetic Abnormality and Associated Genetic Aberration	Epigenetic-Related Change
WDLPS	Ring chromosome 12 12q13-15 region amplifications: *MDM2*, *CDK4*, *HMGA2*, *SAS*, *GL1, JUN* family genes [21,22,23,24,26,27]	Not described
DDLPS	Ring chromosome 12 12q13-15 region amplifications: *MDM2*, *CDK4*, *HMGA2*, *SAS*, *GL1*, *JUN* family genes [21,22,23,24,26,27]	Mutations in genes of epigenetic regulators (HDAC1) Aberrant methylation of tumor-promoting genes KLF4, CEBPA, CDKN2A Increased expression of miR-155 [33,34,35,36]
MLPS	t(12;16) (q13;p11), t(12;22) (q13;q12) *FUS-CHOP*, *EWS-CHOP* [18]	Specific methylation profile of 12q13-q14 region CpG-methylated APC locus and reduced APC expression Epigenetic regulation of increased expression of CDKN2A, MGMT, RASSF1A, MST1, MST2 Increased expression of microRNA-135b [37,38,39,40]
PLPS	13q14.2-5 deletion Rb/TP53 deletion Complex karyotype [21,26,41,42]	Not described
MPLPS	No specific changes Complex karyotype	Not described

**Table 2 cancers-16-00271-t002:** Targeted molecules proposed for LPS treatment.

Drug	Target/Mechanism of Action	LPS Subtype	References
Palbociclib	CDK 4/6 inhibitor	WDLPS, DDLPS	[100,112]
Abemaciclib	CDK 4/6 inhibitor		
Milademetan	*MDM2* inhibitor	WDLPS, DDLPS	[143,144,145,146]
BI 907828 (brigimadlin)	*MDM2* inhibitor		
Sunitinib	PDGFR/VEGFR inhibitor	Metastatic LPS	[98]
Lenvatinib	VEGFR/c-Kit/PDGFR/FGFR/RET inhibitor	LPS	[147]
Pazopanib	PDGFR/VEGFR/FGFR inhibitor	Metastatic LPS	[99]
Efatutazone	PPAR-α inhibitor	MLPS	[125]
Anlotinib	VEGFR/c-Kit/PDGFR/FGFR1 inhibitor	WDLPS/DDLPS	[102,121,122]
Selinexor	Inhibitor of nuclear transportation (inhibitor of exportin 1)	DDLPS	[148]

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
