# Peer review of "Genetic, Epigenetic and Transcriptome Alterations in Liposarcoma for Target Therapy Selection"

_cancers, 2024, doi:10.3390/cancers16020271_

Round 1

Reviewer 1 Report

Comments and Suggestions for Authors

This review is well-organized and educational for the reviewer.  However, the reviewer feels that the following points need to be altered.

1. On the lines 181-182, the authors have written "To date three FUS-CHOP and five EWS-CHOP chimeric genes are described [35]."  However, Powers et al., have shown 11 types of FUS-CHOP variants in Mod Pathol 23: 1307–1315.

2. On the lines 188-195, the authors have referred the mutations in the promoter region of TERT in MRCL. The authors should show the references about this. 

Author Response

We thank the reviewer for the comments to our paper and below provide point-by-point response to them.

1.On the lines 181-182, the authors have written "To date three FUS-CHOP and five EWS-CHOP chimeric genes are described [35]."  However, Powers et al., have shown 11 types of FUS-CHOP variants in Mod Pathol 23: 1307–1315.

We added the correct information on the number of FUS-CHOP chimeric genes as well correct citation (lines 181-182, highlighted in yellow). New reference 49 is also highlighted in yellow in reference list, lines 670-673.

2. On the lines 188-195, the authors have referred the mutations in the promoter region of TERT in MRCL. The authors should show the references about this.

We added the neccessary references [50, 51], line 190, highlighted in yellow, and also added them in reference list, lines 674-679, highlighted in yellow.

Reviewer 2 Report

Comments and Suggestions for Authors

May you discuss the current clinical impact of your researches?

Author Response

We thank the reviewer for the comment and provide the response below.

May you discuss the current clinical impact of your researches?

We summarized here currently developing targeted treatments in liposarcoma research based on genetic and epigenetic features, as well as some clinical trials with the novel molecules. The main clinically relevant conclusion from this summary is the neccessarity of application of combined approach for liposarcoma diagnostics and treatment, including standard radiotherapy, conventional cytostatics, chemoresistance tests ex vivo and in vitro.

Reviewer 3 Report

Comments and Suggestions for Authors

The authors summarized the molecular genetic abnormalities and changes in epigenetic modifiers in liposarcoma to consider their possible therapeutic targets and target therapy drugs proposed for liposarcoma treatment. Finally, they stated that using CDK4/6 inhibitors, MDM2 inhibitors, and multi-tyrosine kinase inhibitors seems promising.

There is much information regarding genetic/epigenetic changes and therapeutic approaches targeted to such changes using drugs in each of the five types of liposarcoma. In the reviewer’s view, he could not catch what the authors wanted to inform the reading audiences about and what the conclusion of the review paper was. There is a conclusion in section 5 but not a conclusion; instead, it seems to be a future perspective. The reviewer asks the authors to revise it so that the reading audience can understand what the crucial points to consider regarding therapeutic targets of chemotherapy for liposarcoma are, what the challenges in the clinical trials are, and how the authors expect the targeted molecular therapy regarding liposarcoma changes the outcome of the cancer.

In addition, the reviewer asks to respond to the following issues.

1. Full spelling of many abbreviations is missing, e.g., YEATS4, FRS2, E2F1, CDKN2A, MIR15A, RUNX3, ARID1A, ATM, CHEK1 (in lines 115-120).

2. Section 2. 4. is not a conclusion of section 2. It may be necessary to add section conclusions in sections 2, 3, and 4 with graphics instead of tables. At present, tables 1 and 2 are not valid summaries because no explanation of the tables exists in the text.

3. Add transcriptome information or change the title if the authors would not add the section for transcriptome.

Author Response

We thank the reviewer for the detailed comments and provide below point-by-point response.

1. There is much information regarding genetic/epigenetic changes and therapeutic approaches targeted to such changes using drugs in each of the five types of liposarcoma. In the reviewer’s view, he could not catch what the authors wanted to inform the reading audiences about and what the conclusion of the review paper was. There is a conclusion in section 5 but not a conclusion; instead, it seems to be a future perspective. The reviewer asks the authors to revise it so that the reading audience can understand what the crucial points to consider regarding therapeutic targets of chemotherapy for liposarcoma are, what the challenges in the clinical trials are, and how the authors expect the targeted molecular therapy regarding liposarcoma changes the outcome of the cancer.

We modified the conclusion according the reviewer comments.

In addition, the reviewer asks to respond to the following issues.

2. Full spelling of many abbreviations is missing, e.g., YEATS4, FRS2, E2F1, CDKN2A, MIR15A, RUNX3, ARID1A, ATM, CHEK 1 (in lines 115-120).

We added the list of abbreviation in the end of the manuscript.

3. Section 2. 4. is not a conclusion of section 2. It may be necessary to add section conclusions in sections 2, 3, and 4 with graphics instead of tables. At present, tables 1 and 2 are not valid summaries because no explanation of the tables exists in the text.

We have revised the conclusions 2.4, 3.3 according to the advice and added the section 4.4 summing up the first attempts of the target therapies of liposarcoma based on molecular studies of the subtype characteristic, therefore, providing the explanation for the table 2 (all changes are highlighted in yellow).

We cited the table1 at the proper places so that it become more clear why we provide the table1

4. Add transcriptome information or change the title if the authors would not add the section for transcriptome.

We describe specific transcriptome alterations as consequences of genetic abnormalities (overexpression and down expression description). Then in the section 4 we described some signaling changes corresponding to genetic alterations in liposarcoma and the development and application of the specific inhibitors for targeted therapy. At the end of the 4th section we added the subsection 4.4, in which we described the first clinical trials of the targeted drugs for liposarcoma treatment and provided the table 2 summing up these data. We would prefer to leave the tables as this type of information presentation seems us more transparent.

Reviewer 4 Report

Comments and Suggestions for Authors

This a well-writtwen paper about the genetic, apigenetic nad trnascriptome alterations in LPS.  The description of molecular genetic,  epigenetic and possible targeted therapies is very thorough and deep. with little practical or clinical consequences. 

Minor: trabectidin is more regarded as chemotherapeutic therapy, the omission of description trabectidin from this paper as targeted drug should be considered.

Some not widely known abbreviations should be solved 8e.g. line 403 "REST")

line 78: the second partof the sentence ("...where MLPS..") a meaning should be given

Author Response

We thank the reviewer for the comments and provide below point-by-point response.

Minor: trabectidin is more regarded as chemotherapeutic therapy, the omission of description trabectidin from this paper as targeted drug should be considered.

Trabectedin is DNA minor groove binder and beyond the interaction with minor groove it was demonstrated the specific interference with several transcription factors, DNA repair pathways as well as the regulation of monocyte and tumor-associated macrophage cell cycle. The described mechanism of trabectedin effects makes this molecule closer to the targeted therapy.

Some not widely known abbreviations should be solved 8e.g. line 403 "REST")

We added the list of abbreviations in the end of the manuscript.

line 78: the second partof the sentence ("...where MLPS..") a meaning should be given

We corrected the sentence.

Reviewer 5 Report

Comments and Suggestions for Authors

Although it is a descriptive review, I think it is a good summary of liposarcoma's genetic, epigenetic, and transcriptomic abnormalities.

On top of that, I have a few opinions.

#1. BI907828 is now given the name brigimadlin (https://pubmed.ncbi.nlm.nih.gov/37269344/). That name should at least be appended.

#2. I feel that several important papers are not included.

- Integrated exome and RNA sequencing of dedifferentiated liposarcoma (https://pubmed.ncbi.nlm.nih.gov/31831742/)

- Myxoid pleomorphic liposarcoma: a clinicopathologic, immunohistochemical, molecular genetic and epigenetic study of 12 cases, suggesting a possible relationship with conventional pleomorphic liposarcoma. (Mod Pathol. 2021 Nov;34(11):2043-2049. doi: 10.1038/s41379-021-00862-2.)

The authors should search the literature again and consider adding any necessary references.

#3. Pleomorphic liposarcoma and myxoid pleomorphic liposarcoma are similar in name but are independent tumor entities and should be discussed in separate sections.

Comments on the Quality of English Language

English is acceptable.

Author Response

We thank the reviewer to the comments and provide below point-by-point response to them.

#1. BI907828 is now given the name brigimadlin (https://pubmed.ncbi.nlm.nih.gov/37269344/). That name should at least be appended

We added this information and the neccessary reference in the Table 2,
hightlighted in yellow, and well as in reference list, highlighted in yellow.

#2. I feel that several important papers are not included.

- Integrated exome and RNA sequencing of dedifferentiated liposarcoma
(https://pubmed.ncbi.nlm.nih.gov/31831742/)

We added the neccessary information and reference on lines 138-139 and
in reference list, all highlighted in yellow.

- Myxoid pleomorphic liposarcoma: a clinicopathologic, immunohistochemical,
molecular genetic and epigenetic study of 12 cases, suggesting a possible
relationship with conventional pleomorphic liposarcoma.
(Mod Pathol. 2021 Nov;34(11):2043-2049. doi: 10.1038/s41379-021-00862-2.)

The authors should search the literature again and consider adding
any necessary references.

We added the neccessary information and reference, lines 91-92 and in the
reference list, highlighted in yellow.

#3. Pleomorphic liposarcoma and myxoid pleomorphic liposarcoma are similar
in name but are independent tumor entities and should be discussed
in separate sections.

We prefer to combine the sections with these liposarcoma subtypes as one of the
main goal of the review was to summarized specific genetic and epigenetic feature, while these subtypes are both characterized by complex karyotype and high heterogeneity.

Round 2

Reviewer 3 Report

Comments and Suggestions for Authors

The revisions mostly answered the major and minor concerns. The only thing is that the full spelling of many abbreviations of the genes is still missing—only the genes I suggested, as some examples, are added to the abbreviation list. The reviewer thinks the manuscript can be acceptable after adding such abbreviations to the list in p13.

Author Response

We updated the list of abbreviations in the end of the manuscript.